# Incidence and temporal changes in lumbar degeneration and low back pain in child and adolescent weightlifters: A prospective 5-year cohort study

Rikuto Yoshimizu[1], Junsuke Nakase[1]*, Katsuhito Yoshioka[2], Kengo Shimozaki[1], Kazuki Asai[1], Mitsuhiro Kimura[1], Katsuhiko Kitaoka[3], Hiroyuki Tsuchiya[1]

1 Department of Orthopedic Surgery, Graduate School of Medical Sciences, Kanazawa University, Kanazawa, Japan, 2 Department of Orthopedic Surgery, National Hospital Organization Kanazawa Medical Center, Kanazawa, Japan, 3 Department of Orthopedic Surgery, Kijima Hospital, Kanazawa, Japan

☯ These authors contributed equally to this work.
* nakase1007@yahoo.co.jp

**Data Availability Statement:** All relevant data are within the paper and its Supporting information files.

## Abstract

This study was conducted to investigate the incidence of lumbar degeneration findings and low back pain (LBP) in children and adolescent weightlifters using magnetic resonance imaging (MRI) and medical questionnaires over a 5-year period. Moreover, we aimed to reveal the temporal changes in the lumbar vertebrae caused by long-term weightlifting training during the growth period. Twelve children and adolescent weightlifters who participated in weightlifting for >2 years (six boys, six girls, 11.4±2.0 years) were enrolled. Participants underwent annual medical questionnaire surveys, including data on practice frequency, competition history, presence of LBP, and lumbar examinations using MRI during the 5-year follow-up. Lumbar disc degeneration was detected in all the participants after 4 years, and lumbar disc herniation findings were detected in 33% of participants after 5 years; one underwent herniotomy during the follow-up period. Lumbar spondylolysis was detected in 58% of patients at 5 years. Although there were three participants who had LBP in the final year, none had LBP that prevented them from returning to weightlifting. This 5-year cohort study of 12 children and adolescent weightlifters detected lumbar degeneration in all participants. High frequency weightlifting training over a long period during the growth period may increase the risk of developing current and future LBP.

## Introduction

Weightlifting training is practiced by athletes of all ages to improve performance and prevent injuries [1]. However, athletes may suffer injuries, such as back pain, if they do not choose the correct weight or have proper form [1, 2]. Particularly, weightlifting training in young athletes can damage the growth plate and should be performed with extreme caution [3]. Existing guidelines emphasize that, for weightlifting training, certified coaches should have appropriate

**Funding:** The authors received no specific funding for this work.

**Competing interests:** The authors have declared that no competing interests exist.

pedagogical experience and communication skills in order to teach young athletes of varying abilities and personalities [4].

Low back pain (LBP) is one of the most common weightlifting complaints, with incidence rates $\geq$ 40%, LBP is caused by lumbar disc degeneration and herniation, which can interfere with not only sports activities but also social life [2]. Although few studies have investigated whether weightlifting training during the growth period is associated with future lumbar disc degeneration and LBP, it is important to note that spinal abnormalities that develop at a young age not only cause LBP at that time, but also increase the risk of recurrence of LBP in adulthood [5, 6]. Therefore, it is necessary to pay close attention to weightlifting training during the growth period.

Previous reports have reported a prevalence of > 80% of radiological changes in the lumbar vertebrae, such as lumbar disc herniation, caused by long-term weightlifting training; however, these reports are from experienced adolescents [7, 8]. Our 3-year prospective cohort study is the only study to observe radiological changes in the lumbar vertebrae of children and adolescents [9]. In this study of 12 child and adolescent weightlifters, 11 participants had detectable lumbar degeneration findings, mainly disc degeneration. It was concluded that resistance training at the competition level in children and adolescents could cause irreversible changes in the lumbar vertebrae.

The purpose of this study was to investigate the incidence of lumbar degeneration and LBP over a 5-year period in child and adolescent weightlifters using magnetic resonance imaging (MRI) and medical questionnaires. This cohort study revealed the temporal changes in the lumbar vertebrae were attributed to long-term weightlifting training during the growth period and have not been revealed so far.

## Methods

This study was conducted between 2014 and 2018, and 12 participants (six boys and six girls) were enrolled (Table 1). The mean age of participants at the start of the study was 11.4 ± 2.0 years; The mean age of the participants at the start of the study was 11.4 ± 2.0 years; the mean body mass index (BMI) was 22.9 kg/m$^2$; and the average competition history was 2.2 ± 0.7 years. Most participants also participated in sports other than weightlifting at the start of the

**Table 1. Participant characteristics.**

| No | Age | Sex | Sports before participation | Sports started during participation |
|----|-----|-----|-----------------------------|-------------------------------------|
| 1 | 14 | boy | Swimming, Sumo | |
| 2 | 13 | boy | Trampoline, Sumo, Judo | |
| 3 | 12 | boy | Swimming, Sumo | |
| 4 | 11 | boy | Swimming | Sumo |
| 5 | 10 | boy | Sumo, Judo | |
| 6 | 8 | boy | Sumo, Judo | Track and field |
| 7 | 14 | girl | Sumo | |
| 8 | 13 | girl | Basketball | |
| 9 | 12 | girl | Basketball, Swimming, Sumo | |
| 10 | 11 | girl | Swimming, Sumo | |
| 11 | 11 | girl | Basketball | |
| 12 | 8 | girl | | |

Prior to starting this study, 11 of the 12 participants participated in non-weightlifting sports, and two participants participated in a new sport in a 5-year follow-up.

study. None of participants had a history of lumbar diseases or surgery and all were followed up for 5 years. The study design was approved by the Ethical Committee of the Graduate School of Medical Sciences, Kanazawa University (approval #1399). The purpose of this study was explained to the participants, and written informed consent was obtained from all participants and their parents. The participants underwent annual medical examinations during the 5-year follow-up period.

In this study, LBP was defined as a condition in which participants were unable to practice weightlifting for more than 1 week due to pain. All the participants adhered to the rule of stopping training when symptoms of LBP appeared and restarted training following the disappearance of the symptoms.

Every year, all participants answered the medical questionnaire regarding the competition history, practice frequency, and presence of LBP each year, and underwent MRI. Each participant was maintained in the supine position with the knee joint in mild flexion during the MRI. MRI of the lumbar vertebrae was performed with a flexible quadrature detection body coil on a 0.4 T unit (APERTO, Hitachi Medical, Tokyo, Japan). T2-weighted images in the sagittal and coronal planes were used to assess the characteristic MRI findings. The section thicknesses of the coronal and sagittal views were 3.5 mm, and the interval gaps for both views were 1.0 mm. We checked for lumbar disc degeneration, disc herniation, and spondylolysis at all lumbar vertebral levels (L1-S1) on MRI scans in the sagittal and coronal planes. Lumbar disc degeneration was assessed using the Pfirrmann classification, which proved adequate agreement among different observers and by the same observer on separate occasions [10, 11]. MRI findings were interpreted independently by two orthopedic surgeons; one was a specialist in spine surgery (reader 1), and the other was an experienced orthopedic surgeon (reader 2). Each surgeon interpreted the MRI findings twice, and the points of each interpretation were separated by a 2-week period. When the judgments of the two surgeons differed, the two readers consulted and adopted the judgment of the spine surgeon. Inter-reader and intra-reader agreements were assessed using κ values [12]. We defined values ≤ 0 as indicating no agreement, 0.01–0.20 as none to slight, 0.21–0.40 as fair, 0.41–0.60 as moderate, 0.61–0.80 as substantial, and 0.81–1.00 as almost perfect agreement.

## Results

The participants practiced approximately 2 hours per day and 5 days per week under the guidance of a team coach. Until the participants reached the age of 10 years, or while the technique was still immature, basic training was conducted to consolidate form and very little training with the barbell was performed. After maturation of their skills, they routinely practiced with weights of approximately 50% of the One Repetition Maximum (1RM), and training with 100% of the 1RM was limited to once a week at most. Prior to starting this study, 11 of the 12 participants participated in non-weightlifting sports, and two participants participated in a new sport during the 5-year follow-up. The mean BMI ranged from 22.9±4.1 kg/m$^2$ to 26.0 ±2.9 kg/m$^2$ over 5 years. No positive findings of lumbar disc herniation and spondylolysis were observed on MRI; LBP was also not observed; however, grade II disc degeneration changes were detected in two participants at the start of this study.

During this 5-year cohort study, eight participants (67%) had lumbar disc degeneration in the second year, nine (75%) in the third year, and 12 (100%) after the fourth year (Table 2). Lumbar disc degeneration was almost irreversible, and the worst grade was as follows: 5 of the 12 cases were grade II, two cases were grade III, and five cases were grade IV in the final year. Grade III or IV was the most common at L4/5 (33%), followed by L5/S (25%), and grade III or IV was detected in two participants at two-disc levels (Table 3). The κ value of inter-reader

**Table 2. Temporal changes of lumbar degeneration findings on MRI and LBP over a 5-year period.**

|  | 2014 | 2015 | 2016 | 2017 | 2018 |
|---|---|---|---|---|---|
| **Lumbar disc degeneration** | 2 | 8 | 9 | 12 | 12 |
| **Lumbar disc herniation** | 0 | 0 | 2 | 3 | 4 |
| **Lumbar spondylolysis** | 0 | 1 | 4 | 1 | 0 |
| **Presence of low back pain** | 0 | 1 | 3 | 2 | 3 |

MRI, magnetic resonance imaging; LBP, lower back pain.

In the final year, abnormal MRI findings were detected in all participants, but only three participants had LBP.

**Table 3. Lumbar disc degeneration grade at each lumbar vertebral level in 2018, as defined by the Pfirrmann classification.**

| Pfirrmann classification | Grade 1 | Grade 2 | Grade 3 | Grade 4 |
|---|---|---|---|---|
| **L1/2** | 6 | 4 | 1 | 1 |
| **L2/3** | 6 | 5 | 1 | 0 |
| **L3/4** | 8 | 3 | 1 | 0 |
| **L4/5** | 4 | 4 | 1 | 3 |
| **L5/S1** | 4 | 5 | 1 | 2 |

Grade III or IV was the most common at L4/5 (33%), followed by L5/S (25%), and grade III or IV was detected in two participants at two-disc levels.

agreement was 0.80 (substantial), and intra-reader agreement was 0.70 (substantial, mean of the readers). Lumbar disc herniation findings were detected in four participants (33%) over 5 years, and one of them underwent herniotomy in 2017.

Lumbar spondylolysis was detected in seven participants (58%) over 5 years, and two of them were found to have it at the same level for 2 consecutive years. It was most common in L5 (33%), followed by L3 (17%). In the final year, abnormal MRI findings were detected in all participants, but only three participants had LBP (Table 2). We present two representative cases of lumbar disc degeneration and spondylolysis, and progressive disc degeneration and disc herniation (Figs 1 and 2).

## Discussion

No prospective studies have focused on lumbar degeneration in child and adolescent weightlifters before epiphyseal closure, other than a previous 3-year cohort study [9]. The present 5-year cohort study revealed that long-term continuation of weightlifting training in children and adolescents is associated with the development of lumbar degeneration, especially disc degeneration. This cohort study will help predict future lumbar degeneration in children and adolescent weightlifters and develop safe training strategies. As mentioned earlier, negative opinions exist on training because weightlifting training in children before epiphyseal closure can lead to growth plate disorders; however, recent studies have demonstrated that proper weight training under the supervision of a qualified adult is effective in improving performance and preventing injuries [1–3]. The participants in this study were competition-level weightlifters; thus, supervisors would have probably provided weightlifters with safe and appropriate guidance regarding training. However, lumbar disc degeneration was detected in all participants after the fourth year, and disc herniation findings were detected in 33% of

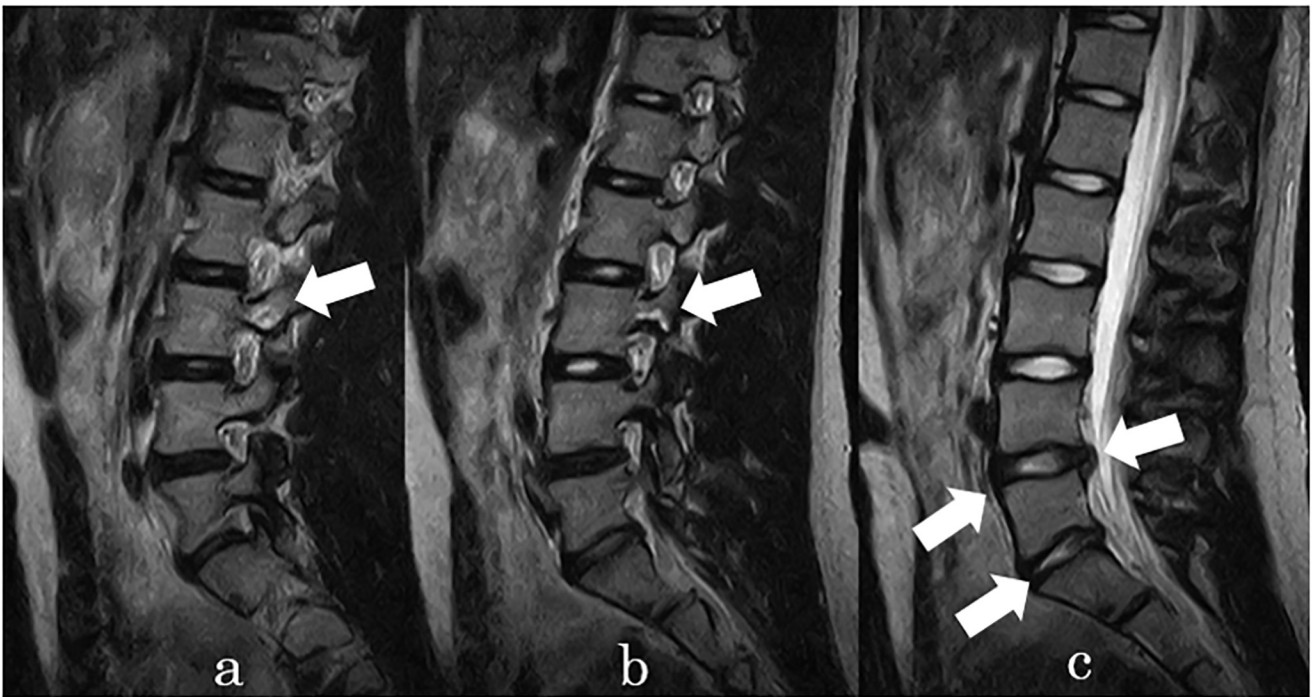

**Fig 1. Representative case (patient 7).** Bilateral lumbar spondylolysis at L3. (a, b) Disc degeneration at L4/5. L5/S1. (c) Disc herniation at L4/5.

participants in the final years; one of the participants underwent herniotomy. Furthermore, lumbar spondylolysis was detected in 58% of the patients; however, none of the participants had chronic LBP or nonunion. These results could indicate that lumbar degeneration occurs frequently with long-term weightlifting training in child and adolescent weightlifters, even under the guidance of supervisors.

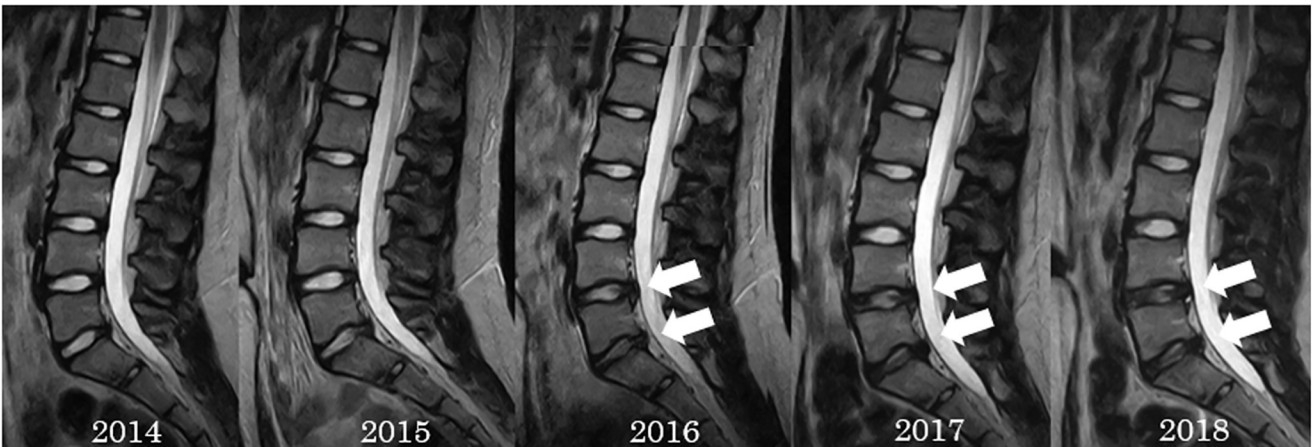

**Fig 2. Representative case (patient 8).** There were no findings in 2014. From 2016, there was progressive lumbar disc degeneration and disc herniation at L4/5 and L5/S1.

Lumbar disc degeneration is characterized by the loss of disc hydration, disc space narrowing, and annular tears [5]. Disc degeneration is considered nearly irreversible since the lumbar disc has limited regenerative capabilities owing to poor vascularity [13], which can lead to other problems, including disc herniation or spondylolysis [14]. Tertti et al. reported that the prevalence of disc degeneration in asymptomatic and symptomatic 15-year-old adolescents was 26% and 38%, respectively [15]. Moreover, Salo et al. reported that the prevalence of disc degeneration in children under the age of 15 years was 22%, and it primarily occurred in children over the age of 10 years [16]. In this study, disc degeneration was detected in $\geq$ 90% of patients in the third year, when the average age of the participants was 15 years. Furthermore, the two youngest 8-year-old children at the beginning of this study had disc degeneration from the age of 9 years.

Lumbar disc herniation is a rare disease in children, and the cumulative incidence of disc herniation below 17 years is 4 in 10,000 [17]. Disc herniation findings were detected in four participants (33%) over 5 years, with a significantly higher incidence. The development of disc degeneration or disc herniation in children is attributed to competition-level sports participation and lifestyle factors [17], and this could be the result of continued stress on disc degeneration. The most important finding of this study was that disc degeneration progressed from a young age in participants without LBP. Disc degeneration is almost irreversible, and participants continue to be at risk of developing LBP in the future [18].

Lumbar spondylolysis is considered a stress fracture owing repetitive hyperextension and axial loading of the spine. It occurs more often in young athletes than in adults, particularly in baseball, gymnastics, football, tennis, and weightlifting [5, 18–20]. In acute spondylolysis, the cure rate is $\geq$ 90% with appropriate conservative treatment. However, nonunion of the fractured part owing to delayed diagnosis or treatment causes chronic LBP, spondylolisthesis, and sciatica [18–21]. Thus, early detection of spondylolysis is crucial to ensure appropriate conservative treatment. T2-weighted MRI is excellent for early indications of spondylolysis, and computed tomography is useful in assessing longitudinal changes in bony union of acute pars defects diagnosed by MRI. Considering these conditions and radiation exposure, MRI is the modality of choice for diagnosing spondylolysis in children. We defined pedicle signal changes as spondylolysis using MRI in this study [5, 20]. In this study, seven cases (58%) of spondylolysis were found in 5 years; however, all were in the early stage. Two of these cases showed early-stage spondylolysis similar to levels in the following year, but none of them developed chronic LBP or nonunion, and spondylolysis was not detected the following year. One case of early-stage spondylolysis was detected in the final year, and the participant to practice following conservative treatment. Spondylolysis is more likely to occur during the growth period [5, 17], which was detected in more than half of the participants in the study. A review of child athletes with LBP reported that exercising 5 days or more than 20 hours a week increased the risk of developing spondylolysis [17], which corresponded with the findings of the participants of this study. Moreover, training more than four times per week does not appear to be any more effective and may increase the risk of overuse injury [22]. On the other hand, the training intensity of the participants in this study was slightly lower than that recommended by the guidelines. This may indicate that not only the type and intensity of training, but also the frequency of training that deviates from the guidelines is a major factor in the development of lumbar degeneration. An appropriate training plan should be selected based on the age range in the guidelines and a coach should supervise the training. Participation in gymnastics, athletics, and court games such as basketball and volleyball are recommended to strengthen the muscles of the whole body, especially when the technique is not yet established [23]. For such athletes, it may also be effective to reduce the load, volume, and frequency of training by incorporating similar training that induces movements that reduce the load on the spine.

This is the first study to prospectively investigate lumbar degeneration in children and adolescent weightlifters. Weightlifting training in children and adolescents requires more careful supervision and long-term follow-up because it frequently causes lumbar degeneration, potentially increasing the risk of developing LBP.

This cohort study had certain limitations. First, because few athletes start weightlifting at the competition levels from childhood or adolescence, the sample size was small. We plan to continue annual medical examinations to evaluate more participants. In addition, no data were available for a control group consisting of children the same age as the participants, so a statistical evaluation was not possible. Second, most participants participated in sports other than weightlifting. Furthermore, two participants started a new sport during the 5-year follow-up period. Therefore, the results of this study may be attributed to the influence of sports other than weightlifting. In particular, during the 5-year observation period, 9 out of 12 participants performed sumo in addition to weightlifting, and the average BMI of the participants during the observation period remained higher than the average for the Japanese population of the same ages. Specifically, the average BMI of Japanese people aged 11.5 years is 17.8±2.4 kg/m$^2$ and 17.6±19 kg/m$^2$ for men and women, respectively [24]. A previous study that observed an association between lumbar disc degeneration and BMI in 16-year-olds reported that the association was found only in males [25]; however, in our study, all 12 participants had disc degeneration at the last observation. In addition, previous studies have reported that various sports, including weightlifting, contribute to lumbar degeneration [5, 20]; however, none of the studies have focused on sumo. Therefore, it is unclear to what extent participation in sumo affects lumbar degeneration; however, since the participants spent most of their time training for weightlifting, we consider that weightlifting training is most likely to be the main factor affecting lumbar degeneration. Third, this study did not compare the participants with controls of similar age groups. Although we do not have unique knowledge of lumbar degeneration in children, the incidence of lumbar degeneration was clearly higher in the participants of this study, based on previous studies [15–17]. Fourth, a mature state was age defined according to the participants' age. Specifically, this study defined the growth period as the school-age period from elementary to junior high school. In particular, participants who were 8 or 14 years old at the start of the study may not have been within their true growth period based on their growth status. Furthermore, it is questionable whether appropriate training prescriptions were strictly administered to participants differences in age and skill. The participants' daily practice was supervised by an experienced coach with a teaching license. On the other hand, the training frequency was higher than that recommended by the guidelines, and the possibility that the training prescription was arbitrary cannot be excluded. Finally, the association between lumbar degeneration detected in the participants and the development of LBP in the future remains unclear, and a longer follow-up period of the participants is required.

In the future, investigating the relationship between lumbar degeneration and LBP over a longer period of time with a larger sample size is warranted. The findings of this study may help to prevent irreversible injuries in children and adolescent athletes undergoing weightlifting training.

## Conclusion

A 5-year cohort study of 12 children and adolescent weightlifters detected lumbar degeneration in all participants. The details of lumbar degeneration were as follows: all (100%) of the 12 participants had disc degeneration, four (33%) participants had disc herniation in the 5-year follow-up; and in seven (58%) participants, spondylolysis was detected by the time of the final

year. High frequency weightlifting training over a long period during the growth period may increase the risk of developing LBP in the future as well as in the present.

## Supporting information

**S1 File. Measurement results.**
(XLSX)

## Acknowledgments

This study would not have been possible without the cooperation of all participants and supportive staff.

## Author Contributions

**Conceptualization:** Junsuke Nakase, Kengo Shimozaki.

**Data curation:** Rikuto Yoshimizu, Kengo Shimozaki, Kazuki Asai, Mitsuhiro Kimura, Katsuhiko Kitaoka.

**Formal analysis:** Rikuto Yoshimizu, Katsuhito Yoshioka, Kengo Shimozaki.

**Investigation:** Rikuto Yoshimizu, Kengo Shimozaki, Kazuki Asai, Mitsuhiro Kimura.

**Methodology:** Rikuto Yoshimizu, Junsuke Nakase, Kengo Shimozaki, Kazuki Asai, Hiroyuki Tsuchiya.

**Project administration:** Junsuke Nakase, Katsuhiko Kitaoka, Hiroyuki Tsuchiya.

**Resources:** Junsuke Nakase.

**Supervision:** Junsuke Nakase, Kengo Shimozaki, Hiroyuki Tsuchiya.

**Validation:** Junsuke Nakase.

**Writing – original draft:** Rikuto Yoshimizu.

**Writing – review & editing:** Junsuke Nakase, Hiroyuki Tsuchiya.

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
