## [Decision Letter · Decision Letter 0]

24 Jan 2022

PONE-D-21-34347Incidence and temporal changes in lumbar degeneration and low back pain in child and adolescent weightlifters: Prospective 5-year cohort studyPLOS ONE

Dear Dr. Nakase,

Thank you for submitting your manuscript to PLOS ONE. After careful consideration, we feel that it has merit but does not fully meet PLOS ONE’s publication criteria as it currently stands. Therefore, we invite you to submit a revised version of the manuscript that addresses the points raised during the review process.

Please address the points that were raised by the reviewers in the body of your manuscript. Information about BMI are essential especially because 9 of 12 participants did sumo wrestling in addition to weight lifting. In addition, please also provide information about a potential correlation between IVD degeneration and sumo wrestling. How much heavier were the participants of this study compared to age matched children/adolescence that did not perform weightlifting and/or sumo wrestling?

We look forward to receiving your revised manuscript.

Kind regards,

Svenja Illien-Jünger, Ph.D.

Academic Editor

PLOS ONE

Journal Requirements:

Reviewers' comments:

Reviewer's Responses to Questions

**Comments to the Author**

1. Is the manuscript technically sound, and do the data support the conclusions?

Reviewer #1: Yes

Reviewer #2: Partly

2. Has the statistical analysis been performed appropriately and rigorously? 

Reviewer #1: N/A

Reviewer #2: No

3. Have the authors made all data underlying the findings in their manuscript fully available?

Reviewer #1: Yes

Reviewer #2: Yes

4. Is the manuscript presented in an intelligible fashion and written in standard English?

Reviewer #1: Yes

Reviewer #2: Yes

5. Review Comments to the Author

Reviewer #1: In this work, the authors explore lumbar degeneration and low back pain in weightlifting adolescents. The manuscript is valuable in that it presents a 5-year prospective study on the back injuries due to weightlifting in children. However, several issues with the manuscript deserve to be raised.

Variables like BMI should be noted as this may play a prominent role in the etiology of back injury. Did the participants follow the same training routines? Please add any possible details that can help show standardization of training across participants.

If possible, knowing which movements or techniques caused acute injuries or exacerbations would be highly valuable.

While the authors did address this in their limitations, the participation of the adolescents in other sports may prominently decrease the validity of the presented findings.

Finally, while the manuscript is generally well written, some areas have minor errors with syntax and readability.

The authors did a fine job with this manuscript. The manuscript provides radiological and clinical follow-up of children and adolescents participating in the sport of weightlifting. Its publication is warranted pending addressing some issues.

Reviewer #2: GENERAL COMMENTS

The authors present a prospective 5- year cohort study investigating the incidence and temporal changes in lumbar degeneration and low back pain in child and adolescent weightlifters. The manuscript has the potential to add to the current understanding of lumbar degeneration and low back pain incidence in young weightlifters, however, I have significant concerns regarding the authors interpretations of the findings from a small sample size, with no control group and limited statistical analysis. For example, it is suggested that weightlifting may increase the risk of developing current and future LBP, however I would argue that this conclusion cannot me made due to the limitations in the study design (i.e., small sample size, no control) and limited statistical analysis. I have provided specific comments below which I hope may help to address these concerns and be useful for future revisions of the manuscript.

SPECIFIC COMMENTS

Abstract - The authors have used the phrase ‘hard weightlifting training’ within the abstract and conclusion. What is meant by this and can a more objective wording be used here to better quantify ‘hard’ i.e., information on training prescription (frequency, volume, intensity)? Related to this, I have concerns regarding the lack of consideration to the training prescription over the 5-year period. Only information on training prescription (duration and frequency) has been presented. Intensity/ loading for example could significantly affect the training adaptations and hence lumbar degeneration yet this has not been mentioned within the manuscript.

Line 34: Can text be added to define the ‘Growth Period’ and outline at what age or maturity status this typically occurs. The authors have not used maturity status to estimate growth status. Therefore, has age been used as a predictor of the growth period? If so, the limitations of this approach should be considered.

Line 41: ‘Particularly, weightlifting training in young athletes can damage the growth plate and should be performed with extreme caution [3].’ Can the authors provide any more information here as to what additional precautions may help to mitigate or reduce these injury risks? E.g., appropriately qualified coaches/ practitioners.

Line 72: Consider replacing ‘our institute’ with institutions name.

Line 126: Text indicates ‘three participants had LBP’ during the final year (2018) however the figure indicates two participants. Ensure this is consistent.

Line 161: The participants took part in weightlifting training 2hrs/ day, 5 days a week. This is referred to as ‘appropriate prescription’. However, given existing guidelines on training frequency and duration (e.g., Lloyd et al., 2012) it could be argued that this prescription is above the recommendation for the participants’ age and maturity status. The results therefore may be a result of the high training exposure, rather than the type of training/ weightlifting alone. I strongly believe this should be considered in the interpretation to prevent any misconceptions from readers.

Lloyd, R. S., Oliver, J. L., Meyers, R. W., Moody, J. A., & Stone, M. H. (2012). Long-term athletic development and its application to youth weightlifting. Strength & Conditioning Journal, 34(4), 55-66.

Line 187: Missing reference for ‘Disc degeneration is almost irreversible, and participants continue to be at risk of developing LBP in the future.’

Line 212: ‘Proper conservative treatment’ is recommended as an effective tool to prevent ‘irreversible disability’. Can the authors make any recommendations based on the study to prevent the injury occurrence in the first instance based on the study findings or existing research? E.g., incorporating movements that reduce spinal loading but elicit similar training adaptations into training and reduced load, volume, frequency.

Line 220: It is mentioned that a study limitation is the participants' participation in additional sports outside of weightlifting. The influence of concurrent sports training on the study findings is considered. However, the authors should also consider recommendations against early specialisation in a single sport (for increased performance and reduced injury risk). Early specialisation could also be considered in line 184, where it is mentioned ‘disc herniation in children is attributed to competition-level sports participation and lifestyle factors'.

Line 233- It is suggested that ‘the findings of this study may help to prevent irreversible injuries in children and adolescent athletes undergoing weightlifting training.’ Throughout the manuscript however, there are very few recommendations on specifically how these LB injuries can be prevented. This information needs to be added to strengthen the studies application. Furthermore, based on the small sample size and lack of statistical analysis it may be appropriate to temper these interpretations. Finally, it would be advised that statistical analysis (in addition to agreement) is conducted.

6. PLOS authors have the option to publish the peer review history of their article (what does this mean?). If published, this will include your full peer review and any attached files.

Reviewer #1: No

Reviewer #2: No

---

## [Author Response · Author response to Decision Letter 0]

22 Mar 2022

PONE-D-21-34347

Incidence and temporal changes in lumbar degeneration and low back pain in child and adolescent weightlifters: Prospective 5-year cohort study

PLOS ONE

Dear Dr. Nakase,

Thank you for submitting your manuscript to PLOS ONE. After careful consideration, we feel that it has merit but does not fully meet PLOS ONE’s publication criteria as it currently stands. Therefore, we invite you to submit a revised version of the manuscript that addresses the points raised during the review process.

Please address the points that were raised by the reviewers in the body of your manuscript. Information about BMI are essential especially because 9 of 12 participants did sumo wrestling in addition to weightlifting. In addition, please also provide information about a potential correlation between IVD degeneration and sumo wrestling. How much heavier were the participants of this study compared to　age matched children/adolescence that did not perform weightlifting and/or sumo wrestling?

→ Thank you for pointing this out. I have added the information on the BMI of the participants and the Japanese population of the same age (Line 88-91, Line 137-138, Line 263-270). However, I could not find any study that investigated the relationship between sumo and lumbar degeneration in children. This could be due to the remarkably small population of the study participants.

A rebuttal letter that responds to each point raised by the academic editor and reviewer(s). You should upload this letter as a separate file labeled 'Response to Reviewers'. A marked-up copy of your manuscript that highlights changes made to the original version. You should upload this as a separate file labeled 'Revised Manuscript with Track Changes'. An unmarked version of your revised paper without tracked changes. You should upload this as a separate file labeled 'Manuscript'. If applicable, we recommend that you deposit your laboratory protocols in protocols.io to enhance the reproducibility of your results. Protocols.io assigns your protocol its own identifier (DOI) so that it can be cited independently in the future. For instructions see: https://journals.plos.org/plosone/s/submission-guidelines#loc-laboratory-protocols. Additionally, PLOS ONE offers an option for publishing peer-reviewed Lab Protocol articles, which describe protocols hosted on protocols.io. Read more information on sharing protocols at https://plos.org/protocols?utm_medium=editorial-email&utm_source=authorletters&utm_campaign=protocols.

→ As you pointed out, I have created a laboratory protocol, and the DOI is below; dx.doi.org/10.17504/protocols.io.b5vqq65w

We look forward to receiving your revised manuscript.

Kind regards,

Svenja Illien-Jünger, Ph.D.

Academic Editor

PLOS ONE

Journal Requirements: When submitting your revision, we need you to address these additional requirements.1. Please ensure that your manuscript meets PLOS ONE's style requirements, including those for file naming. The PLOS ONE style templates can be found athttps://journals.plos.org/plosone/s/file?id=wjVg/PLOSOne_formatting_sample_main_body.pdf andhttps://journals.plos.org/plosone/s/file?id=ba62/PLOSOne_formatting_sample_title_authors_affiliations.pdf2. In your Data Availability statement, you have not specified where the minimal data set underlying the results described in your manuscript can be found. PLOS defines a study's minimal data set as the underlying data used to reach the conclusions drawn in the manuscript and any additional data required to replicate the reported study findings in their entirety. All PLOS journals require that the minimal data set be made fully available. For more information about our data policy, please see http://journals.plos.org/plosone/s/data-availability. Upon re-submitting your revised manuscript, please upload your study’s minimal underlying data set as either Supporting Information files or to a stable, public repository and include the relevant URLs, DOIs, or accession numbers within your revised cover letter. For a list of acceptable repositories, please see http://journals.plos.org/plosone/s/data-availability#loc-recommended-repositories. Any potentially identifying patient information must be fully anonymized. Important: If there are ethical or legal restrictions to sharing your data publicly, please explain these restrictions in detail. Please see our guidelines for more information on what we consider unacceptable restrictions to publicly sharing data: http://journals.plos.org/plosone/s/data-availability#loc-unacceptable-data-access-restrictions. Note that it is not acceptable for the authors to be the sole named individuals responsible for ensuring data access. We will update your Data Availability statement to reflect the information you provide in your cover letter.

→ Thank you for pointing this out. I have uploaded the data as a Supporting Information file.

Reviewers' comments:

Reviewer's Responses to Questions Comments to the Author

1.Is the manuscript technically sound, and do the data support the conclusions?

The manuscript must describe a technically sound piece of scientific research with data that supports the conclusions. Experiments must have been conducted rigorously, with appropriate controls, replication, and sample sizes. The conclusions must be drawn appropriately based on the data presented. Reviewer #1: Yes Reviewer #2: Partly2. Has the statistical analysis been performed appropriately and rigorously? Reviewer #1: N/A Reviewer #2: No3. Have the authors made all data underlying the findings in their manuscript fully available?

The PLOS Data policy requires authors to make all data underlying the findings described in their manuscript fully available without restriction, with rare exception (please refer to the Data Availability Statement in the manuscript PDF file). The data should be provided as part of the manuscript or its supporting information, or deposited to a public repository. For example, in addition to summary statistics, the data points behind means, medians and variance measures should be available. If there are restrictions on publicly sharing data—e.g. participant privacy or use of data from a third party—those must be specified. Reviewer #1: Yes Reviewer #2: Yes4. Is the manuscript presented in an intelligible fashion and written in standard English?

PLOS ONE does not copyedit accepted manuscripts, so the language in submitted articles must be clear, correct, and unambiguous. Any typographical or grammatical errors should be corrected at revision, so please note any specific errors here. Reviewer #1: YesReviewer#2: Yes5. Review Comments to the Author

Please use the space provided to explain your answers to the questions above. You may also include additional comments for the author, including concerns about dual publication, research ethics, or publication ethics. (Please upload your review as an attachment if it exceeds 20,000 characters) Reviewer #1: In this work, the authors explore lumbar degeneration and low back pain in weightlifting adolescents. The manuscript is valuable in that it presents a 5-year prospective study on the back injuries due to weightlifting in children. However, several issues with the manuscript deserve to be raised.

Variables like BMI should be noted as this may play a prominent role in the etiology of back injury. Did the participants follow the same training routines? Please add any possible details that can help show standardization of training across participants.

If possible, knowing which movements or techniques caused acute injuries or exacerbations would be highly valuable.

→ Thank you for pointing this out. We have added a note on the BMI of the participants (Line 88-91, Line 137-138, Line 263-270). We have also added the details about the training to the results (Line130-136).

While the authors did address this in their limitations, the participation of the adolescents in other sports may prominently decrease the validity of the presented findings.

→ We agree with your point. Unfortunately, no paper demonstrated an association between sumo and lumbar degeneration; however, the athletes spent most of their club time training for weightlifting. We have added these details to the research limitations (Line 260-276).

Finally, while the manuscript is generally well written, some areas have minor errors with syntax and readability.

→ Thank you for pointing this out. The manuscript has been revised for syntax and readability.

The authors did a fine job with this manuscript. The manuscript provides radiological and clinical follow-up of children and adolescents participating in the sport of weightlifting. Its publication is warranted pending addressing some issues.

Reviewer #2: GENERAL COMMENTS

The authors present a prospective 5- year cohort study investigating the incidence and temporal changes in lumbar degeneration and low back pain in child and adolescent weightlifters. The manuscript has the potential to add to the current understanding of lumbar degeneration and low back pain incidence in young weightlifters, however, I have significant concerns regarding the authors interpretations of the findings from a small sample size, with no control group and limited statistical analysis. For example, it is suggested that weightlifting may increase the risk of developing current and future LBP, however I would argue that this conclusion cannot me made due to the limitations in the study design (i.e., small sample size, no control) and limited statistical analysis. I have provided specific comments below which I hope may help to address these concerns and be useful for future revisions of the manuscript.

SPECIFIC COMMENTS

Abstract - The authors have used the phrase ‘hard weightlifting training’ within the abstract and conclusion. What is meant by this and can a more objective wording be used here to better quantify ‘hard’ i.e., information on training prescription (frequency, volume, intensity)? Related to this, I have concerns regarding the lack of consideration to the training prescription over the 5-year period. Only information on training prescription (duration and frequency) has been presented. Intensity/ loading for example could significantly affect the training adaptations and hence lumbar degeneration yet this has not been mentioned within the manuscript.

→ Thank you for pointing this out. The ambiguous word "hard" has been removed throughout the text. We have added as much information as possible in the text about the training that the participants were undergoing (Line130-136). The abstract has been revised, mentioning in the discussion that the participants training exceeded the guidelines’ recommendations (Line238-248).

Line 34: Can text be added to define the ‘Growth Period’ and outline at what age or maturity status this typically occurs. The authors have not used maturity status to estimate growth status. Therefore, has age been used as a predictor of the growth period? If so, the limitations of this approach should be considered.

→ Thank you for pointing this out. In this study, school-age children from elementary school to junior high school were defined as being in the ‘growth period’. Maturity status was not used to estimate growth status. As you pointed out, this is a limitation of this study, and we have discussed it in the text (Line 279-282) 

Line 41: ‘Particularly, weightlifting training in young athletes can damage the growth plate and should be performed with extreme caution [3].’ Can the authors provide any more information here as to what additional precautions may help to mitigate or reduce these injury risks? E.g., appropriately qualified coaches/ practitioners.

→ Thank you for pointing this out. As you have mentioned, much of the literature emphasizes training limitations and supervision by a licensed coach until the proper form is acquired. We have added this information to our discussion (Line 61-63).

Line 72: Consider replacing ‘our institute’ with institutions name.

→ Thank you for pointing this out. We have made the necessary revisions (Line 93-95).

Line 126: Text indicates ‘three participants had LBP’ during the final year (2018) however the figure indicates two participants. Ensure this is consistent.

→ Thank you for pointing this out. We have corrected the table accordingly (Line 155 Table2).

Line 161: The participants took part in weightlifting training 2hrs/ day, 5 days a week. This is referred to as ‘appropriate prescription’. However, given existing guidelines on training frequency and duration (e.g., Lloyd et al., 2012) it could be argued that this prescription is above the recommendation for the participants’ age and maturity status. The results therefore may be a result of the high training exposure, rather than the type of training/ weightlifting alone. I strongly believe this should be considered in the interpretation to prevent any misconceptions from readers.

Lloyd,R. S., Oliver, J. L., Meyers, R. W., Moody, J. A., & Stone, M. H. (2012). Long-term athletic development and its application to youth weightlifting. Strength & Conditioning Journal, 34(4), 55-66.

→　Thank you for providing the references. As you have accurately pointed out, the results were much higher than the frequency of practice recommended in the guidelines. We have revised the results and discussion to include more details regarding the training (Line 130-136, Line 238-252).

Line 187: Missing reference for ‘Disc degeneration is almost irreversible, and participants continue to be at risk of developing LBP in the future.’

→ Thank you for pointing this out. We have added the corresponding reference (Line 219). 

Line 212: ‘Proper conservative treatment’ is recommended as an effective tool to prevent ‘irreversible disability’. Can the authors make any recommendations based on the study to prevent the injury occurrence in the first instance based on the study findings or existing research? E.g., incorporating movements that reduce spinal loading but elicit similar training adaptations into training and reduced load, volume, frequency.

→ Thank you for your suggestions and ideas. As mentioned above, we have added the information you suggested in addition to training in compliance with the guidelines and proper supervision of the coach (Line 246-252).

Line 220: It is mentioned that a study limitation is the participants' participation in additional sports outside of weightlifting.The influence of concurrent sports training on the study findings is considered. However, the authors should also consider recommendations against early specialisation in a single sport (for increased performance and reduced injury risk). Early specialisation could also be considered in line 184, where it is mentioned ‘disc herniation in children is attributed to competition-level sports participation and lifestyle factors'.

→ Thank you for pointing this out. As you have mentioned, participation in sports other than weightlifting is recommended for a while after starting weightlifting or while the technique is not yet established. I have included and elaborated on your ideas in the discussion (Line 248-250).

Line 233- It is suggested that the findings of this study may help to prevent irreversible injuries in children and adolescent athletes undergoing weightlifting training.’ Throughout the manuscript however, there are very few recommendations on specifically how these LB injuries can be prevented. This information needs to be added to strengthen the studies application. Furthermore, based on the small sample size and lack of statistical analysis it may be appropriate to temper these interpretations. Finally, it would be advised that statistical analysis (in addition to agreement) is conducted.

→ Thank you for pointing this out. We have added the information　about injury prevention based on this study (Line 246-252). We have also corrected the exaggerated expression in the last sentence (Line 287-289). Small sample size and no control group, statistical analysis was added to the limitations (Line 257-261).

6. PLOS authors have the option to publish the peer review history of their article (what does this mean?). If published, this will include your full peer review and any attached files.

→ I have no problem with review information being made public.

Do you want your identity to be public for this peer review? For information about this choice, including consent withdrawal, please see our Privacy Policy.Reviewer #1: NoReviewer #2: No

→ I uploaded two files. Both comments were "Please inspect this version for image clarity and content. Is this correct?

---

## [Decision Letter · Decision Letter 1]

20 Apr 2022

PONE-D-21-34347R1Incidence and temporal changes in lumbar degeneration and low back pain in child and adolescent weightlifters: A prospective 5-year cohort studyPLOS ONE

Dear Dr. Nakase,

Thank you for submitting your manuscript to PLOS ONE. After careful consideration, we feel that it has merit but does not fully meet PLOS ONE’s publication criteria as it currently stands. Therefore, we invite you to submit a revised version of the manuscript that addresses the points raised during the review process.

We look forward to receiving your revised manuscript.

Kind regards,

Svenja Illien-Jünger, Ph.D.

Academic Editor

PLOS ONE

Journal Requirements:

Reviewers' comments:

Reviewer's Responses to Questions

**Comments to the Author**

1. If the authors have adequately addressed your comments raised in a previous round of review and you feel that this manuscript is now acceptable for publication, you may indicate that here to bypass the “Comments to the Author” section, enter your conflict of interest statement in the “Confidential to Editor” section, and submit your "Accept" recommendation.

Reviewer #1: All comments have been addressed

Reviewer #2: (No Response)

2. Is the manuscript technically sound, and do the data support the conclusions?

Reviewer #1: Yes

Reviewer #2: Partly

3. Has the statistical analysis been performed appropriately and rigorously? 

Reviewer #1: Yes

Reviewer #2: No

4. Have the authors made all data underlying the findings in their manuscript fully available?

Reviewer #1: Yes

Reviewer #2: Yes

5. Is the manuscript presented in an intelligible fashion and written in standard English?

Reviewer #1: Yes

Reviewer #2: Yes

6. Review Comments to the Author

Reviewer #1: The authors adequately addressed my previous comments. While the studies does possess some flaws, these were appropriately addressed in the limitations section. I recommend acceptance as the positive aspects of the study outweigh potential flaws.

Reviewer #2: The authors have shown a good attempt to address all previous comments and concerns with the manuscript. However, additional concerns are still present, as outlined below.

Line 63- Consider changing to ‘weightlifting training’ rather than S&C.

Line 97- Remove ‘no’ prior to history.

Line 134- Edits suggest that statistical analysis was conducted (in SPSS), however it is mentioned in the limitations that this was not possible (line 280). I would argue the lack of statistical analysis is still downplayed, despite being a major limitation of the study.

Line 142- Consider changing to ‘One Repetition Maximum (1RM)’ to ensure clarity. The prescription of these loads seems rather arbitrary. Are you able to provide a reference to support this prescription?

Line 306- The wording here is a little unclear. Consider referring to the fact that chronological age was used to assume maturity status. It would be useful to reference research supporting the limitations of this approach. Also, additional important information could be noted with regards to the training. Was technique assured throughout and were the sessions supervised by a qualified personal?

7. PLOS authors have the option to publish the peer review history of their article (what does this mean?). If published, this will include your full peer review and any attached files.

Reviewer #1: No

Reviewer #2: No

---

## [Author Response · Author response to Decision Letter 1]

24 May 2022

Reviewer #1: The authors adequately addressed my previous comments. While the studies does possess some flaws, these were appropriately addressed in the limitations section. I recommend acceptance as the positive aspects of the study outweigh potential flaws.

→ Thank you very much for reviewing our manuscript.

Reviewer #2: The authors have shown a good attempt to address all previous comments and concerns with the manuscript. However, additional concerns are still present, as outlined below.

Line 63- Consider changing to ‘weightlifting training’ rather than S&C.

→ Thank you for pointing this out. We have corrected the text (Line 61).

Line 97- Remove ‘no’ prior to history.

→ Thank you for pointing this out. We have corrected the text (Line 92).

Line 134- Edits suggest that statistical analysis was conducted (in SPSS), however it is mentioned in the limitations that this was not possible (line 280). I would argue the lack of statistical analysis is still downplayed, despite being a major limitation of the study.

→ As you pointed out, we believe that the most significant limitation of this study is the lack of statistical analyses. We have removed the sentence about using SPSS (Line 126).

Line 142- Consider changing to ‘One Repetition Maximum (1RM)’ to ensure clarity. The prescription of these loads seems rather arbitrary. Are you able to provide a reference to support this prescription?

→ We have made the required revisions in accordance with your suggestion (Line 133-134). As mentioned in the Discussion section, the training load on the participants is likely to be lower than that recommended by the guidelines, while the training frequency is likely to be higher. We were unable to find any literature to support such a training prescription and, as you note, we must acknowledge that it was arbitrary. We have addressed this issue in the revised Limitations section (Line 284-289).

Line 306- The wording here is a little unclear. Consider referring to the fact that chronological age was used to assume maturity status. It would be useful to reference research supporting the limitations of this approach. Also, additional important information could be noted with regards to the training. Was technique assured throughout and were the sessions supervised by a qualified personal?

→ We have revised the text in accordance with your comments. We have also added information on the training prescription and supervision, based on the aforementioned information (Line 280-284).

---

## [Editor Report · Decision Letter 2]

3 Jun 2022

Incidence and temporal changes in lumbar degeneration and low back pain in child and adolescent weightlifters: A prospective 5-year cohort study

PONE-D-21-34347R2

Dear Dr. Nakase,

We’re pleased to inform you that your manuscript has been judged scientifically suitable for publication and will be formally accepted for publication once it meets all outstanding technical requirements.

Kind regards,

Svenja Illien-Jünger, Ph.D.

Academic Editor

PLOS ONE

---

## [Editor Report · Acceptance letter]

20 Jun 2022

PONE-D-21-34347R2 

Incidence and temporal changes in lumbar degeneration and low back pain in child and adolescent weightlifters: A prospective 5-year cohort study 

Dear Dr. Nakase:

I'm pleased to inform you that your manuscript has been deemed suitable for publication in PLOS ONE. Congratulations! Your manuscript is now with our production department. 

Kind regards, 

on behalf of

Dr. Svenja Illien-Jünger 

Academic Editor

PLOS ONE